# Tensile Behavior Characteristics of High-Performance Slurry-Infiltrated Fiber-Reinforced Cementitious Composite with Respect to Fiber Volume Fraction

**DOI:** 10.3390/ma12203335

**Published:** 2019-10-13

**Authors:** Seungwon Kim, Dong Joo Kim, Sung-Wook Kim, Cheolwoo Park

**Affiliations:** 1Department of Civil Engineering, Kangwon National University, 346 Jungang-ro, Samcheok 25913, Korea; 2Department of Civil and Environmental Engineering, Sejong University, 209 Neungdong-ro, Gwangjin-gu, Seoul 05006, Korea; djkim75@sejong.ac.kr; 3Department of Infrastructure Safety Research, Korea Institute of Civil Engineering and Building Technology 283, Goyang-daero, Ilsanseo-gu, Goyang 10223, Korea; swkim@kict.re.kr

**Keywords:** SIFRCC, fiber volume fraction, direct tensile strength, energy absorption capacity, direct tensile test

## Abstract

Concrete has high compressive strength, but low tensile strength, bending strength, toughness, low resistance to cracking, and brittle fracture characteristics. To overcome these problems, fiber-reinforced concrete, in which the strength of concrete is improved by inserting fibers, is being used. Recently, high-performance fiber-reinforced cementitious composites (HPFRCCs) have been extensively researched. The disadvantages of conventional concrete such as low tensile stress, strain capacity, and energy absorption capacity, have been overcome using HPFRCCs, but they have a weakness in that the fiber reinforcement has only 2% fiber volume fraction. In this study, slurry infiltrated fiber reinforced cementitious composites (SIFRCCs), which can maximize the fiber volume fraction (up to 8%), was developed, and an experimental study on the tensile behavior of SIFRCCs with varying fiber volume fractions (4%, 5%, and 6%) was carried out through direct tensile tests. The results showed that the specimen with high fiber volume fraction exhibited high direct tensile strength and improved brittleness. As per the results, the direct tensile strength is approximately 15.5 MPa, and the energy absorption capacity was excellent. Furthermore, the bridging effect of steel fibers induced strain hardening behavior and multiple cracks, which increased the direct tensile strength and energy absorption capacity.

## 1. Introduction

Concrete is widely used in architectural structures and social infrastructure facilities because it is economical and has high compressive strength and durability. However, concrete is characterized by brittle fracture due to low bending and tensile strengths and weak crack resistance compared to the high compressive strength [1,2,3,4,5,6]. Recently, studies have been conducted to develop high-performance construction materials with excellent performance by improving the disadvantages or maximizing the advantages of concrete. Attempts are being made to mechanically increase strength or improve ductility, because strength and ductility have opposite properties [7]. 

With the development of construction technology, the construction of high-rise of buildings and long structures, the use of 100 MPa or higher ultra-high-strength concrete is increasing [8]. However, although ultra-high-strength concrete has high compressive strength, it has low tensile strength, bending strength, and toughness, and weak resistance to cracks [8]. Above all, ultra-high-strength concrete has the intrinsic problem of brittle fracture to the peak stress. To overcome this problem, active studies are being conducted on high-performance fiber-reinforced cementitious composites (HPFRCCs), in which brittle fracture to ductile fracture are induced [8].

HPFRCCs are characterized by improved tensile force, strain capacity, and energy absorption capacity, which are the weaknesses of conventional concrete [9,10]. However, the maximum fiber volume fraction of conventional HPFRCCs and fiber-reinforced concrete is limited to 2.0% due to fiber balling [9,10]. A majority of fiber balling occurs during the fiber addition process due to excessive fibers during the mixing of HPFRCCCs and fiber-reinforced concrete. Thus, the limited fiber volume fraction has been one of the biggest disadvantages. Studies are being actively conducted to understand the direct tensile behavior characteristics of HPFRCCs [9,10]. To overcome these disadvantages, an increase in the steel fiber volume fraction may improve tensile strength and energy absorption capacity. Therefore, HPFRCCs reported thus far have exhibited deflection hardening behavior under flexural tensile load, rather than direct tensile behavior and strain hardening behavior with multiple micro-cracks [9,10]. Therefore, it is very difficult to obtain a tensile stress-strain curve under direct tensile load to acquire information on multiple cracks [9,10].

To maximize the mechanical properties of HPFRCCs and overcome the limitation of fiber volume fraction, this study developed slurry-infiltrated fiber-reinforced cementitious composites (SIFRCCs), which can incorporate a high volume of steel fibers. The SIFRCCs can incorporate up to 8% fiber volume fraction, thus maximizing tensile strength, energy absorption capacity, and strain capacity, which are shortcomings of the conventional concrete and fiber-reinforced concrete. An experimental research on the tensile behavior characteristics was conducted with respect to the fiber volume fraction of high-performance SIFRCCs through a direct tensile test. 

## 2. Existing Works Related to Direct Tensile Test

In a study on the flexural tensile strength of fiber-reinforced concrete members, the ductile behavior improved after cracking at 80 MPa or lower compressive strength. However, studies on structural behavior analysis of members of 150 MPa or higher compressive strength are relatively insufficient and predictions of the bending strength are limited [8,11]. Flexural tensile tests of fiber-reinforced concrete involve many difficulties, but flexural tensile test are mainly conducted for direct tensile tests because of the problems of slip phenomenon of specimens in the drawing process [8]. However, attempts at direct tensile tests are being made continuously because reliability can be compromised due to many assumption conditions in the process of estimating tensile strength through flexural tensile tests. To address this problem, French regulations have presented a method of performing tests by directly installing notches in the specimens [12].

According to a research report on ultra high performance concrete (UHPC), crack review appears to be unnecessary for UHPC considering its higher tensile strength than conventional concrete due to the action of steel fibers, and the characteristic of low crack width relative to the tensile load also appears to be unnecessary [8,13,14]. The report mentioned that even though the crack width is small, it is necessary to examine crack behavior under various load conditions [8,15]. It also stated that because the crack behavior can vary under the working load and extreme load due to the ductile tensile behavior, it is necessary to validate the crack examination of UHPC with the existing design standards through direct tensile test [8,15]. Furthermore, unlike the direct tensile test, indirect tensile tests, such as tensile test through bending, are burdensome because they require inverse analysis using numerical analysis [8,15]. Although direct tensile tests are not widely used in the measurement of the tensile behavior of concrete, they have been considered to be the most direct and proper method owing to the characteristics of UHPC that exhibits ductile behavior after cracking [8,15].

## 3. Experiment Overview

### 3.1. SIFRCCs

The SIFRCCs developed in this study can maximize the steel fiber volume fraction of the existing fiber-reinforced concrete and HPFRCCs. With its high fiber volume ratio, high tensile strength, energy absorption capacity, and strain capacity can be expected. Unlike fiber-reinforced concrete and HPFRCCs, SIFRCCs can be produced by the following steps. First, the mold is filled with steel fibers in advance. Second, high-performance slurry is prepared after mixing. This slurry should be poured to avoid mixing the concrete matrix and steel fibers. The high-performance slurry should be pour from one end to the other within the cluster of steel fibers such that there are no bubbles. This is done to avoid voids. SIFRCCs are characterized by the omission of coarse aggregates for the high-performance slurry to achieve sufficient filling performance between the steel fibers [9,16].

### 3.2. Experiment Method

To analyze the compressive strength of SIFRCCs, fiber volume fractions of 4%, 5%, and 6% were considered. The compressive strength test was performed in accordance with Korea Standards (KS F 2405) [17]. The cylinder specimens used had a diameter of 100 mm and a height of 200 mm.

To analyze the tensile behavior characteristics with respect to the fiber volume fraction of SIFRCCs, the characteristics of direct tensile behavior were experimented. In the case of direct tensile test, the test method is not clearly defined. Therefore, we conducted an experimental study on the tensile behavior characteristics such as energy absorption capacity through the direct tensile strength, which indicates the maximum tensile stress, strain capacity at the direct tensile strength, and stress-strain curve based on literature review.

To analyze the tensile behavior characteristics of SIFRCCs, fiber volume fractions of 4%, 5%, and 6% were considered. The direct tensile test was performed by displacement control method at the rate of 1 mm/min. A direct tensile test specimen appropriate for the dedicated tensile jig was fabricated as shown in Figure 1. The tensile behavior characteristics were analyzed using the 300-ton class universal testing machine shown in Figure 2. The cross-section of the direct tensile test specimen has a width of 50 mm and a height of 25 mm. The scope of gauge length for tensile performance measurement was set to 50 mm. To induce multiple cracks of SIFRCCs within the range of gauge length, a wire mesh was installed outside the gauge length and used facilitate crack inducement. Furthermore, to derive the tensile stress-strain curve of the SIFRCCs, a linear variable differential transformer (LVDT) (Tokyo Sokki, Tokyo, Japan) that can take measurements up to 25 mm was installed on either side of the tensile jig. The direct tensile strength was calculated using the following Equation (1) [7]:(1)$$f=\frac{P_{max}}{bh}$$
where $P_{max}$ is the maximum load (N), b is the width (mm) at the gauge length, and h. is the height at the gauge length (mm). 

### 3.3. Materials

This study used type 1 ordinary Portland cement. Table 1 lists the physical and chemical properties of the used cement. In this study, silica fume was used to achieve high-performance and high strength of the slurry. Table 2 lists the physical and chemical properties of the used silica fume.

Furthermore, fine aggregates with a diameter of 0.5 mm or less were used to improve the filling performance of the high-performance slurry and to reduce material separation. Coarse aggregates were not used to secure filling performance. To improve the filling performance of the slurry, a high-performance polycarboxylic acid water-reducing agent was used. The admixture used in this experiment has high strength and high flow characteristics and has excellent unit water quantity reduction property and material separation resistance. 

For steel fibers, double hook steel fibers for conventional concrete with a diameter of 0.75 mm, a length of 60 mm, and an aspect ratio of 80 were used. Regarding physical properties, the steel fibers have a density of 7.8 g/cm^3^ and a tensile strength of 1200 MPa. Figure 3 shows the shape of the used steel fibers.

### 3.4. Mixing and Fabrication of Specimens

To mix the SIFRCCs, the water-binder ratio was fixed to 0.35 to achieve the filling performance of the high-performance slurry for filling the inner space of the steel fibers that were placed in advance. The amount of the high range water reducing (HRWR) agent was set to 2.5% of the binder weight. To reduce material separation and achieve the required strength, fine aggregates were added for 0.5% of binder weight and the silica fume was added for 15% of the cement weight. Table 3 shows the SIFRCC mixing formula. The fiber volume fraction variables were set to 4%, 5%, and 6%.

To analyze the tensile behavior characteristics of the SIFRCCs with respect to the fiber volume fraction of 4%, 5%, and 6% in the direct tensile test, direct tensile test specimens were fabricated with the mixing ratio of each variable in Table 3. 

## 4. Results and Analysis

### 4.1. Compressive Strength

Figure 4 shows the compressive strength test results with respect to the fiber volume fraction of SIFRCCs. In the case of 6% fiber volume fraction, the average compressive strength was analyzed to be approximately 83 MPa. The average compressive strength of 5% fiber volume fraction was approximately 75 MPa, lower by approximately 10%. Furthermore, the average compressive strength of 4% fiber volume fraction was approximately 66 MPa lower by approximately 12% compared to the 5% fiber volume fraction and by approximately 21% compared to the 6% fiber volume fraction. The compressive strength increased in proportion to the fiber volume fraction. The input amount of steel fibers appeared to increase with increasing fiber volume fraction, which generated the restraining effect of the specimen itself, and this affected the increase of compressive strength. Figure 5 shows the result of the compressive stress-strain test with respect to the fiber volume fraction.

### 4.2. Direct Tensile Strength

Figure 6 shows the results of the direct tensile strength test with respect to the fiber volume fraction of SIFRCCs. The direct tensile strength test result of 6% fiber volume fraction showed a high average direct tensile strength of approximately 15.5 MPa. The average direct tensile strength of 5% fiber volume fraction was approximately 14.2 MPa, lower by approximately 9%. The average direct tensile strength of 4% fiber volume fraction was approximately 11.0 MPa, lower by approximately 23% compared to 5% fiber volume fraction and by approximately 30% compared to 6% fiber volume fraction.

The direct tensile strength test of the SIFRCCs showed that the cracks gradually spread and lead to fracture after the initial cracking due to the reinforcement of steel fibers. This phenomenon was evident as the fiber volume fraction increased. Furthermore, the direct tensile strength also showed an increasing trend with the increasing fiber volume fraction, similar to the compressive strength test result of the SIFRCCs with respect to the fiber volume fraction.

### 4.3. Strain Capacity and Tensile Stress-Strain Curve

Figure 7 shows the strain capacity test result with respect to the fiber volume fraction at the direct tensile strength. The result of the strain capacity test under the direct tensile strength with respect to the fiber volume fraction of SIFRCCs verified excellent strain capacity of 0.7% (0.007) at 5% fiber volume fraction. Since the direct tensile strength increased with increasing fiber volume fraction, the strain capacity was expected increase as well, but the strain capacity was the lowest at 6% fiber volume fraction. This is because in the case of 6% fiber volume fraction, the steel fibers resist the direct tensile load as the fiber amount increases, and due to the small cross-section size (25 × 50 mm^2^) of the direct tensile test specimen, the adhesion performance of the high-performance slurry matrix and steel fibers decreased. Furthermore, considering that the length of the steel fibers is 60 mm, the size of the specimens is considered to be affected by the specimens because the arrangement of steel fibers was parallel to the tensile load when the direct tensile test specimen was fabricated.

Figure 8 shows the tensile stress-strain curve with respect to the fiber volume fraction of the SIFRCCs, and Figure 9 shows the tensile stress-strain curve of the 6% fiber volume fraction of the SIFRCCs. The strain at the direct tensile strength was analyzed to be 0.53% (0.0053), and the energy absorption capacity was 62.10 kJ/m^3^, which was the lowest among all variables. Regarding the compressive stress-strain test result for 6% fiber volume fraction, the post-peak behavior exhibited a strain hardening behavior, but the post-peak behavior of the tensile stress-strain curve test result exhibited a strain softening behavior. Considering that the cross-section size of the gauge length for measurement is 25 × 50 mm^2^, the result was somewhat different from the compression behavior because the adhesion performance between the high-performance slurry matrix and steel fibers was insufficient.

Figure 10 shows the tensile stress-strain curve of the SIFRCCs with 5% fiber volume fraction. The strain at the direct tensile strength was 0.7% (0.0070), indicating the highest strain capacity among all variables. The energy absorption capacity was also the highest at 88.05 kJ/m^3^ on average. Similar to the tensile stress-strain test result for the specimen with 6% fiber volume fraction, the post-peak behavior exhibited a strain softening behavior.

Figure 11 shows the tensile stress-strain curve of SIFRCCs with 4% fiber volume fraction. Unlike the tensile stress-strain test result for 5% and 6% fiber volume fractions, the post-peak behavior characteristics were closest to the strain hardening behavior. The direct tensile strength for 4% fiber volume fraction was lower than those of others, but the energy absorption capacity was 62.27 kJ/m^3^, which is higher than that of the 6% fiber volume fraction (the highest direct tensile strength). This is considered to be because the size of the direct tensile test specimen is too small and the interface adhesion property between the high-performance slurry and steel fibers did not reach the maximum, resulting in different tensile behavior characteristics for each fraction.

## 5. Conclusions

To overcome the limited fiber volume fraction of the conventional fiber-reinforced concrete and HPFRCCs, this study developed SIFRCCs that contains a high fiber volume fraction to maximize the tensile strength, energy absorption capacity and strain capacity. An experimental study on tensile behavior characteristics was conducted with respect to different fiber volume fractions of the high-performance SIFRCCs through direct tensile tests. The conclusions of this study are as follows.

(1)The analysis result of the tensile behavior characteristics through direct tensile tests of the SIFRCCs showed a high direct tensile strength, more than 15 MPa (Vf = 6%), which is higher than conventional HPFRCCs and UHPC due to increasing the fiber volume fraction. Also the load of SIFRCCs with respect to the fiber volume fraction continuously increased because of the high fiber volume fraction after the initial crack, and sufficient residual strength was obtained after the maximum strength. This sufficient residual strength is expected to bring about positive effects to the brittle fracture of structures when unexpected loads is applied.(2)The reinforcement with a high fiber volume fraction improved brittleness, which is a disadvantage of conventional concrete. After the initial cracking, cracks gradually spread and led to fracture. The maximum strain capacity was approximately 0.7%, which showed excellent energy absorption capacity. However, the interface adhesion performance between the high-performance slurry and steel fibers was insufficient due to small cross-section of the direct tensile test specimen. It is expected that the strain capacity and energy absorption capacity can be improved by increasing the cross-section size.(3)The energy absorption capacity increased but the strain capacity tended to decrease with increasing fiber volume fractions. It is that the size of the direct tensile specimen is too small to exert the maximum interface adhesive characteristics between the high-performance slurry and steel fibers, indicating different direct tensile behavior for each fiber volume fraction.(4)The bridging effect of steel fibers caused strain hardening behavior and multiple cracks, resulting in the increase of the direct tensile strength and energy absorption capacity. In the case of the strain capacity, it was suppressed by the high-performance slurry restraint due to the increase of the adhesion of the slurry and steel fibers.

## Figures and Tables

**Figure 1 materials-12-03335-f001:**
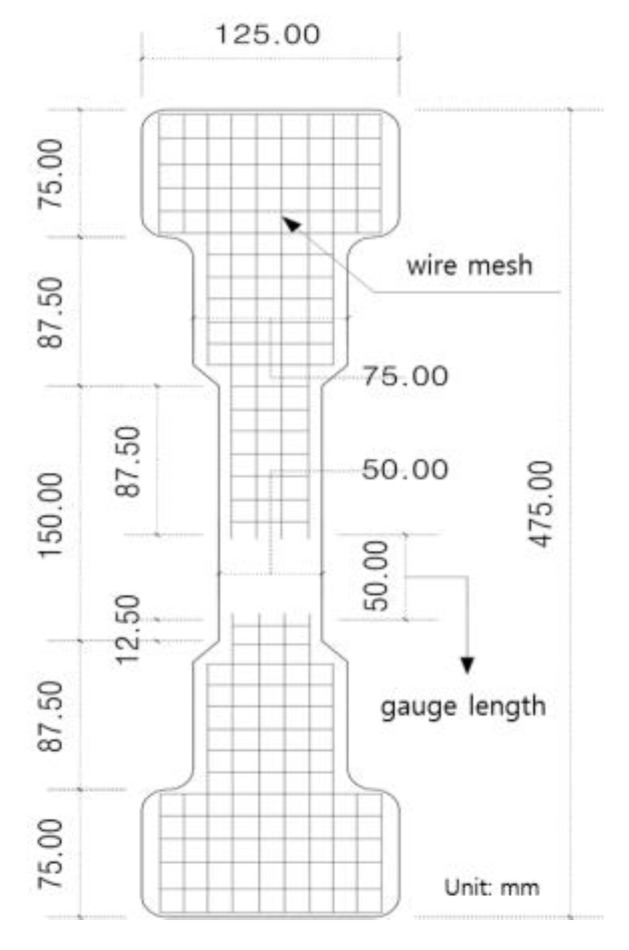
Schematic of the specimen.

**Figure 2 materials-12-03335-f002:**
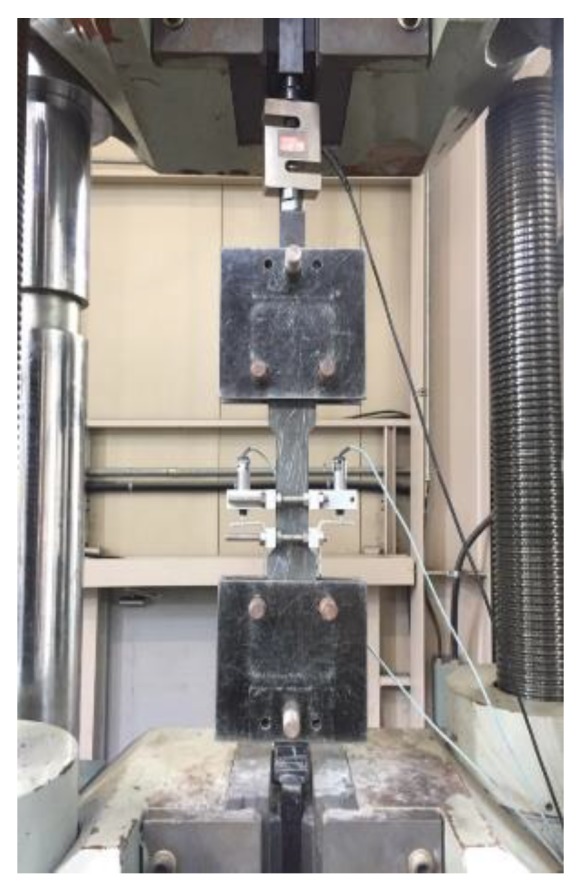
Experimental setup.

**Figure 3 materials-12-03335-f003:**
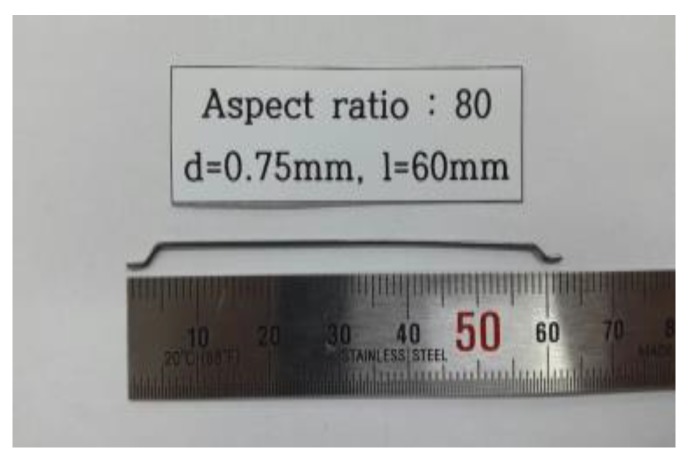
Shape of the used steel fibers.

**Figure 4 materials-12-03335-f004:**
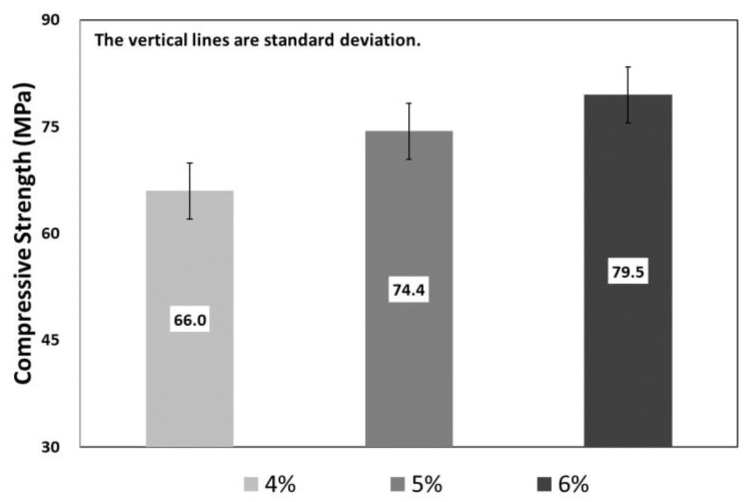
Compressive strength test results with respect to the fiber volume fraction.

**Figure 5 materials-12-03335-f005:**
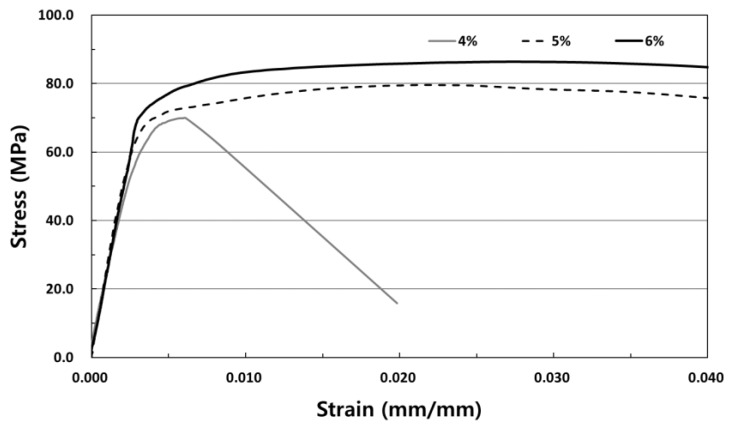
Compressive stress-strain curve with respect to the fiber volume fraction.

**Figure 6 materials-12-03335-f006:**
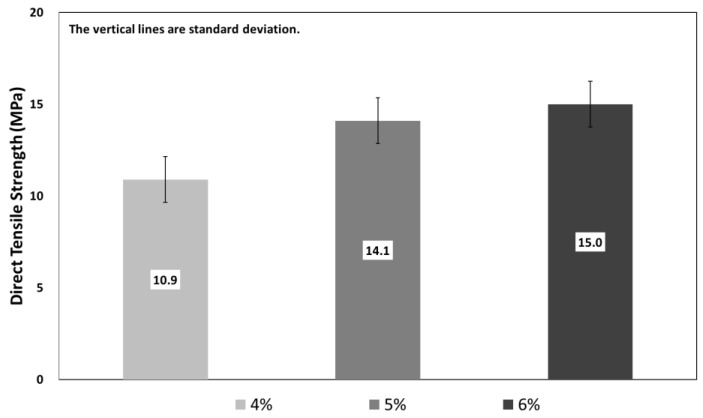
Direct tensile strength with respect to fiber volume fraction.

**Figure 7 materials-12-03335-f007:**
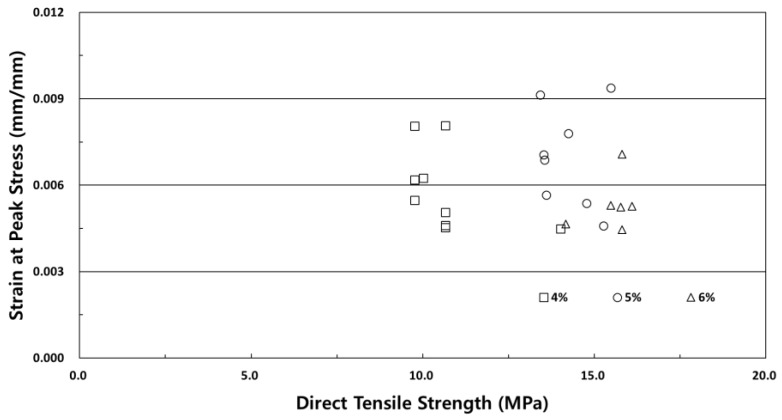
Strain capacity test results at the direct tensile strength.

**Figure 8 materials-12-03335-f008:**
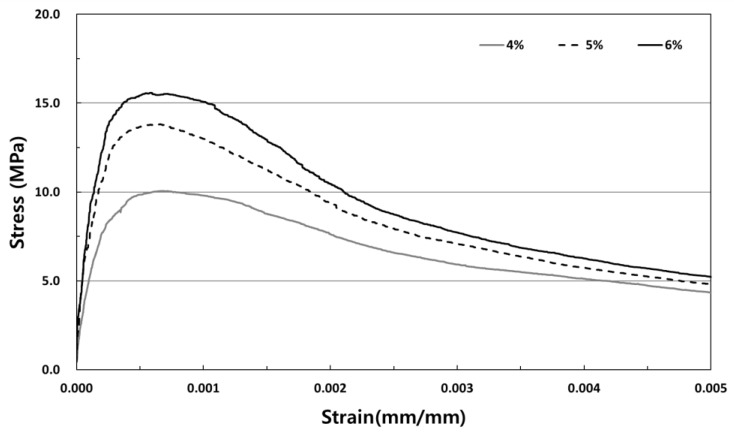
Tensile stress-strain curve with respect to the fiber volume fraction.

**Figure 9 materials-12-03335-f009:**
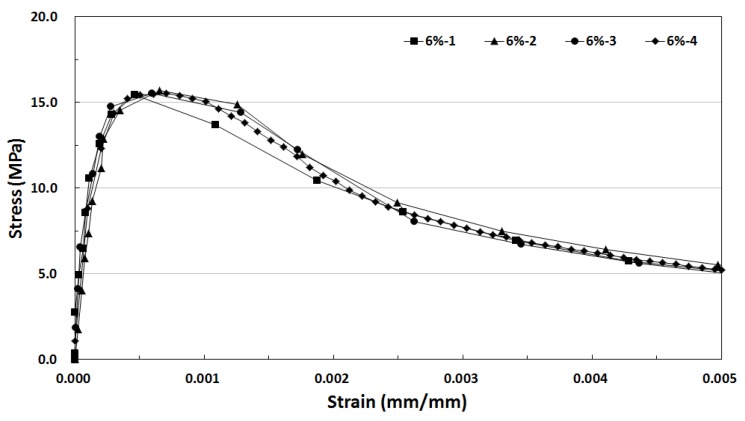
Tensile stress-strain curve of Slurry-Infiltrated Fiber-Reinforced Cementitious Composites (SIFRCCs) with 6% fiber volume fraction.

**Figure 10 materials-12-03335-f010:**
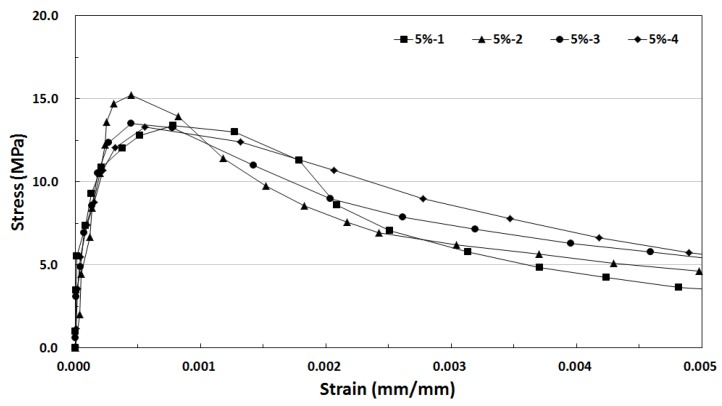
Tensile stress-strain curve of SIFRCCs with 5% fiber volume fraction.

**Figure 11 materials-12-03335-f011:**
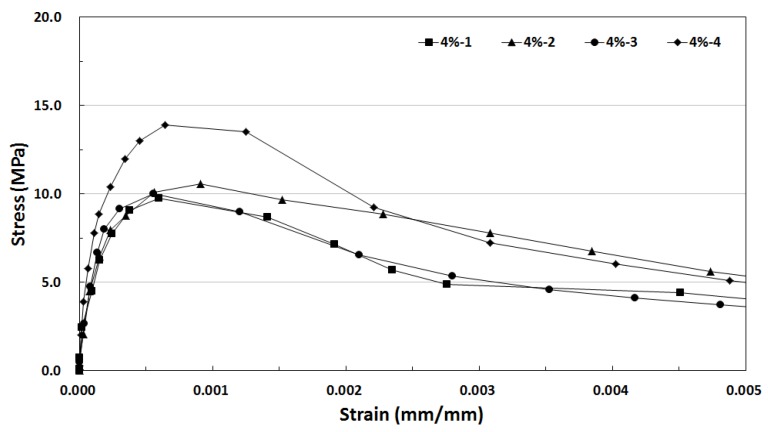
Tensile stress-strain curve of SIFRCCs with 4% fiber volume fraction.

**Table 1 materials-12-03335-t001:** Physical and chemical properties of the used cement.

Physical Properties							
**Specific Gravity**	**Fineness (cm^2^/g)**		**Stability (%)**	**Setting Time (min)**			**LOI (%)**
				**Initial**		**Final**	
3.15	3400		0.10	230		410	2.58
**Chemical compositions (%, mass)**							
SiO_2_	CaO	MgO	SO_3_		Al_2_O_3_		
21.95	60.12	3.32	2.11		6.59		

**Table 2 materials-12-03335-t002:** Physical and chemical properties of silica fume.

Physical Properties					
**Specific Gravity**			**Fineness (cm^2^/g)**		
2.10			200,000		
**Chemical compositions (%, mass)**					
SiO_2_	CaO	MgO		SO_3_	Al_2_O_3_
96.00	0.38	0.10		-	0.25

**Table 3 materials-12-03335-t003:** Slurry-Infiltrated Fiber-Reinforced Cementitious Composites (SIFRCCs) mixing formula.

Variables	W/B (%)	Unit Material Quantity (kg/m^3^)					
		W	C	Fine Aggregate	Silica Fume	HRWR	Steel Fibers
4%	35	407.4	962.8	566.4	169.9	28.3	312
5%							390
6%							468

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
