# Peer review of "Tensile Behavior Characteristics of High-Performance Slurry-Infiltrated Fiber-Reinforced Cementitious Composite with Respect to Fiber Volume Fraction"

_materials, 2019, doi:10.3390/ma12203335_

Round 1
Reviewer 1 Report
The present work developed experimental tests on Slurry Infiltrated Fiber Reinforced Cementitious Composites (SIFRCCs), which can incorporate a high volume of steel fibers with respect to HPFRCCs.
Specific comments:
Line 36-39: Concrete is widely used in architectural structures and social infrastructure facilities because it is economical and has high compressive strength and durability. However, concrete is characterized by brittle fracture due to low bending and tensile strengths and weak crack resistance compared to the high compressive strength [1-6]. The beginning of Introduction Section starts with a well-known consideration and many references (6) are referring to it. In the reviewer's opinion, a deeper description of the CFRP system and the HPFRCC one is needed, also provinding the advantages and disadvantages of using one or the other one. In this framework, for instance, the paper "Ultrasonic inspection for the detection of debonding in CFRP-reinforced concrete" has to be cited. Line 53: please provide explanation of the fiber ball phenomenon. Line 96-97: SIFRCCs can be produced by first dispersing and placing in advance with steel fibers and then filling high-performance slurry, rather than mixing the concrete matrix and steel fibers together. This is sentence is not clear. Please, provide more explanations (and even better clarified) about how SIFRCCs are produced. By looking at Eq.(1), it is not clear why the authors refers to post-cracking strength even though it is calculated with the ultimate load. It should be named ultimate strength. Line 252: As the cracks gradually spread and led to fracture after the
initial cracking. This sentence is maybe is complete.
Author Response
Response to Reviewer 1 Comments
Point 1:
Line 36-39: Concrete is widely used in architectural structures and social infrastructure facilities because it is economical and has high compressive strength and durability. However, concrete is characterized by brittle fracture due to low bending and tensile strengths and weak crack resistance compared to the high compressive strength [1-6]. The beginning of Introduction Section starts with a well-known consideration and many references (6) are referring to it. In the reviewer's opinion, a deeper description of the CFRP system and the HPFRCC one is needed, also provinding the advantages and disadvantages of using one or the other one. In this framework, for instance, the paper "Ultrasonic inspection for the detection of debonding in CFRP-reinforced concrete" has to be cited.
Response 1:
Thank you for your comments.
The content of the CFRP system is not use for this paper.
Point 2:
Line 53: please provide explanation of the fiber ball phenomenon.
Response 2:
Thank you for your comments.
Information on fiber balling has been added in the manuscript.
Point 3:
Line 96-97: SIFRCCs can be produced by first dispersing and placing in advance with steel fibers and then filling high-performance slurry, rather than mixing the concrete matrix and steel fibers together. This is sentence is not clear. Please, provide more explanations (and even better clarified) about how SIFRCCs are produced. By looking at Eq.(1), it is not clear why the authors refers to post-cracking strength even though it is calculated with the ultimate load. It should be named ultimate strength.
Response 3:
Thank you for your comments.
Information on SIFRCCs has been added in the manuscript. In addition, a sentence that is not clear are clearly corrected.
Post-cracking strength is converted to ‘direct tensile strength’.
Point 4:
Line 252: As the cracks gradually spread and led to fracture after the initial cracking. This sentence is maybe complete.
Response 4:
Thank you for your comments.
The sentence is corrected.

Reviewer 2 Report
The manuscript presents a very interesting topic and concerns the assessment of Cementitious Composite properties. High tensile strength is an important element in the design of modern cement concrete. The subject matter is within the scope of the journal. The methodology is sufficiently well explained that someone else knowledgeable about the field could repeat the study. Each figure and table is necessary to the understanding of the conclusions. All elements of the manuscript relate logically to the study's statement of purpose. The work is well written but needs some adjustments. As a conclusion of this manuscript is acceptable for publication after the major revision.
Some suggestions for improvement are given as follows:
Abstract:
- The abstract need to be summarised the main points and avoid the unnecessary parts to better understanding and readability.
-Please underscore the scientific value added in the abstract. Add some of the most critical quantitative results to the Abstract.
Introduction:
- There is no discussion in the introduction about other ways to improve the tensile strength of cement concrete.
- Please describe the most important results of previously published studies
- Several footnotes to one general sentence are unacceptable (eg line 39 [1-6]). Please provide the results, conclusions from the research presented in the literature.
Experiment Overview:
- The methodology is sufficiently well explained that someone else knowledgeable about the field could repeat the study although subject is quite detailed and complicated.
- Figure 1 - illegible, please correct the dimensions.
- Figure 4 - standard samples, drawing is of no scientific value, please delete.
Results and Analysis
- Results and Analysis should be improved and supplemented with detailed analysis and discussion. In this section you should not only discuss the results of the research but also discuss them in comparison to the results presented in the literature. You must explain why these results were obtained.
- Fig. 5 - 13- illegible, please correct.
- Fig. 5, 7 - what do the vertical lines mean? Standard deviation? This must be explained and described.
Conclusion
The conclusions relate to conducted research, they do not require improvement.

Author Response
Response to Reviewer 2 Comments
Point 1:
Abstract:
- The abstract need to be summarised the main points and avoid the unnecessary parts to better understanding and readability.
-Please underscore the scientific value added in the abstract. Add some of the most critical quantitative results to the Abstract.
Response 1:
Thank you for your comments.
The abstract has been summarised and quantitative data added to the abstract.
Point 2:
Introduction:
- There is no discussion in the introduction about other ways to improve the tensile strength of cement concrete.
- Please describe the most important results of previously published studies
- Several footnotes to one general sentence are unacceptable (eg line 39 [1-6]). Please provide the results, conclusions from the research presented in the literature.
Response 2:
Thank you for your comments.
The introduction part has been corrected.
We are allowed to use several citation according to ‘Reference List and Citations Style Guide for MDPI Journals’.
Point 3:
Experiment Overview:
- The methodology is sufficiently well explained that someone else knowledgeable about the field could repeat the study although subject is quite detailed and complicated.
- Figure 1 - illegible, please correct the dimensions.
- Figure 4 - standard samples, drawing is of no scientific value, please delete.
Response 3:
Thank you for your comments.
The experiment overview part has been corrected.
Point 4:
Results and Analysis
- Results and Analysis should be improved and supplemented with detailed analysis and discussion. In this section you should not only discuss the results of the research but also discuss them in comparison to the results presented in the literature. You must explain why these results were obtained.
- Fig. 5 - 13- illegible, please correct.
- Fig. 5, 7 - what do the vertical lines mean? Standard deviation? This must be explained and described.
Response 4:
Thank you for your comments.
Results and analysis has been improved and supplemented with detailed analysis and discussion.
Figures are corrected and vertical lines are standard daviation.
Point 5:
Conclusion
The conclusions relate to conducted research, they do not require improvement.
Response 5:
Thank you for your comments.

Reviewer 3 Report
This paper is too poor to be published. The English would need complete rewriting. The investigated fiber reinforced material is compared with concrete but it is an ultra fine mortar. Figure 3 is unacceptable. The caption of Fig. 4 is misleading. It is not explained how compressive tests were run. What is the meaning of post cracking strength? just to mention a number of weak points.
Author Response
Response to Reviewer 3 Comments
Point 1:
This paper is too poor to be published. The English would need complete rewriting. The investigated fiber reinforced material is compared with concrete but it is an ultra fine mortar. Figure 3 is unacceptable. The caption of Fig. 4 is misleading. It is not explained how compressive tests were run. What is the meaning of post cracking strength? just to mention a number of weak points.
Response 1:
Thank you for your comments.
In the case of UHPC, it is the ultra fine mortar case that does not use coarse aggregate, but uses only fine aggregate. No coarse aggregate was used in this study, which can be compared to UHPC.
The steel fibers used in this study, figure 3, were purchased and used produced from steel fibers company. So there may be few errors in the production process.
Figure 4 was deleted.
The compression test was performed in accordance with KS standards (KS F 2405). The compression tests were explained in the manuscript.
The post-cracking strength means direct tensile strength. Post-cracking strength is changed to ‘direct tensile strength’.

Round 2
Reviewer 2 Report
The article has been amended in accordance with the comments made in the review and can be published.
Author Response
Thank you for your comments.

Reviewer 3 Report
This paper should not be published as it is. The English language is too poor and often misleading.The new lines 98 - 101 are an example. Terms like "at home and abroad" (see line 103 p.e.) cannot be used in a serious paper. The shape of the specimen is not dog bone. On each page you will find at least five sentences, which would need rewriting. The conclusions would also need rewriting. This already starts with the first sentence.
Author Response
Thank you for your comments.
We have modified the contents of the overall paper on the areas of misleading.
In addition, we reviewed the terms “at home and abroad”, deleted and revised them in the paper.
The dog-bone replaced with tensile test specimen.
On each page, we checked and modified correctly.
The conclusions have been modified and rewrote.
Through the English editing service, the paper was revised overall.
Thank you again for the review.
